# Insufficient Sun Exposure Has Become a Real Public Health Problem

**DOI:** 10.3390/ijerph17145014

**Published:** 2020-07-13

**Authors:** Lars Alfredsson, Bruce K. Armstrong, D. Allan Butterfield, Rajiv Chowdhury, Frank R. de Gruijl, Martin Feelisch, Cedric F. Garland, Prue H. Hart, David G. Hoel, Ramune Jacobsen, Pelle G. Lindqvist, David J. Llewellyn, Henning Tiemeier, Richard B. Weller, Antony R. Young

**Affiliations:** 1Institute of Environmental Medicine, Karolinska Institute, 171 77 Stockholm, Sweden; Lars.Alfredsson@ki.se; 2School of Population and Global Health, The University of Western Australia, Perth 6009, Australia; bruce@brucekarmstrong.org; 3Department of Chemistry and Sanders-Brown Center on Aging, University of Kentucky, Lexington, KY 40506, USA; dabcns@uky.edu; 4Cardiovascular Epidemiology Unit, Department of Public Health and Primary Care, University of Cambridge, Cambridge CB1 8RN, UK; RC436@medschl.cam.ac.uk; 5Department of Dermatology, Leiden University Medical Centre, 2333 ZA Leiden, The Netherlands; growl.frank@gmail.com; 6Clinical & Experimental Sciences, University of Southampton Medical School and University Hospital Southampton NHS Foundation Trust, Southampton SO16 6YD, UK; M.Feelisch@soton.ac.uk; 7Division of Epidemiology, Department of Family Medicine and Public Health, University of California San Diego School of Medicine, La Jolla, CA 92093, USA; cgarland@ucsd.edu; 8Telethon Kids Institute, University of Western Australia, Perth 6872, Australia; Prue.Hart@telethonkids.org.au; 9Department of Public Health Sciences, College of Medicine, Medical University of South Carolina, Charleston, SC 29425, USA; 10Department of Pharmacy, University of Copenhagen, DK-2100 Copenhagen, Denmark; ramune.jacobsen@sund.ku.dk; 11Department of Clinical Science and Education, Karolinska Institute, 171 77 Stockholm, Sweden; pelle.lindqvist@ki.se; 12College of Medicine and Health, University of Exeter Medical School, Exeter EX1 2LU, UK; david.llewellyn@exeter.ac.uk; 13Department of Social and Behavioral Science, Harvard T.H. Chan School of Public Health, Harvard University, Cambridge, MA 02115, USA; tiemeier@hsph.harvard.edu; 14Centre for Inflammation Research, University of Edinburgh, Edinburgh EH16 4SB, UK; richard.weller@ed.ac.uk; 15St John’s Institute of Dermatology, King’s College London, London SE1 9RT, UK; antony.young@kcl.ac.uk

**Keywords:** disease prevention, UV radiation, sun exposure, vitamin D, nitric oxide

## Abstract

This article aims to alert the medical community and public health authorities to accumulating evidence on health benefits from sun exposure, which suggests that insufficient sun exposure is a significant public health problem. Studies in the past decade indicate that insufficient sun exposure may be responsible for 340,000 deaths in the United States and 480,000 deaths in Europe per year, and an increased incidence of breast cancer, colorectal cancer, hypertension, cardiovascular disease, metabolic syndrome, multiple sclerosis, Alzheimer’s disease, autism, asthma, type 1 diabetes and myopia. Vitamin D has long been considered the principal mediator of beneficial effects of sun exposure. However, oral vitamin D supplementation has not been convincingly shown to prevent the above conditions; thus, serum 25(OH)D as an indicator of vitamin D status may be a proxy for and not a mediator of beneficial effects of sun exposure. New candidate mechanisms include the release of nitric oxide from the skin and direct effects of ultraviolet radiation (UVR) on peripheral blood cells. Collectively, this evidence indicates it would be wise for people living outside the tropics to ensure they expose their skin sufficiently to the sun. To minimize the harms of excessive sun exposure, great care must be taken to avoid sunburn, and sun exposure during high ambient UVR seasons should be obtained incrementally at not more than 5–30 min a day (depending on skin type and UV index), in season-appropriate clothing and with eyes closed or protected by sunglasses that filter UVR.

## 1. Background

Over the past century, work has largely migrated from outdoors to indoors. Digital attractions and air conditioning have resulted in people spending more recreational time sheltered from the natural environment in which humans evolved. A major environmental factor is solar terrestrial ultraviolet radiation (UVR wavelengths from ~295 to 400 nm). This spectral region can subdivided into UVB radiation (wavelengths 280–315 nm) and UVA radiation (wavelengths 315–400 nm), of which the majority (>95%) is UVA. The intensity (irradiance, W/m^2^) of UVB is very sensitive to the height of the sun, which depends on latitude, season and time of day: the higher the sun, the greater the UVB content of sunlight, which also means that the ratio of UVB:UVA is constantly changing.

Solar radiation has many effects on human health, all of which are underpinned by molecular (e.g., DNA damage) and cellular changes (e.g., Langerhans and dermal dendritic cell migration). Most research to date has focused on adverse effects, with vitamin D synthesis being seen as the only established benefit. Sunburn (erythema) is the most obvious acute clinical effect and skin cancer is the most important chronic effect. UVR dose (J/m^2^ = W/m^2^ × time (s)) received by viable skin is an important factor, and this depends on many biological (e.g., amount of melanin), behavioral (e.g., clothing and sunscreen cover) and environmental factors (e.g., latitude), all of which influence the exposure ratio to ambient (ERTA). This is also dependent on angle of the sun on the body site in question (e.g., horizonal vs. vertical surface) [1]. Different photobiological endpoints in the skin and eye have different UVR dose thresholds, spectral dependencies and exposure patterns in the case of chronic outcomes. A very important public health tool for estimating solar UVR irradiance is the ultraviolet index (UVI), which is an indicator of erythemal intensity under any given circumstance.

Reduction of time outdoors has been amplified by skin cancer prevention campaigns to minimize sun exposure (e.g., by the US Centers for Disease Control and Prevention (CDC)) [2]. While it is accepted that solar UVR exposure is the main cause of skin cancer [3], evidence is accumulating on the health benefits of sun exposure [3], as well as widespread vitamin D deficiency [3,4], and has revealed a possibly significant public health problem resulting from insufficient sun exposure. This suggests that current public health advice on sun exposure ought to be reconsidered to communicate a better balance of the benefits and harms of sunlight, particularly at higher latitudes where ambient levels of UVR are comparatively low even in summer.

On May 6–7, 2019, the authors met in Washington, D.C., to discuss research on sun exposure and human health. The narrative that follows is a mini non-systematic review that summarizes the main discussions at this meeting, which included some of the world’s foremost experts on the health effects of sun exposure.

Because the cutaneous production of vitamin D is dependent on solar UVB exposure, this vitamin (in fact a hormone) has commonly been assumed to account for all the beneficial effects of sunshine that have been suggested by ecologic studies (e.g., latitude gradients of some diseases). More recently, there has been greater awareness of the diversity of cutaneous mediators released in response to UVR, which also has a large effect on the skin transcriptome [5,6]. There is also evidence that solar UVR may affect the blood transcriptome [6,7], including genes for immunity. This suggests that reported beneficial effects of sunlight may be mediated by multiple signalizing molecules, individually or in concert, rather than via a single mediator such as vitamin D. Hence, in the following discussion, higher vitamin D concentrations (measured as serum 25-hydroxy vitamin D (25(OH)D)) are considered as a proxy for sun exposure so that their association with health benefits will be taken as indicating a beneficial effect of sunlight but not necessarily a benefit of vitamin D.

## 2. Evidence for Health Benefits of Sun Exposure

### 2.1. All-Cause Mortality

Chowdhury et al. [8] studied serum 25(OH)D data from 849,000 participants and concluded that 12.8% of all US deaths (340,000/year) and 9.4% of all deaths in Europe (480,000/year) could be attributed to serum 25(OH)D <75 nmol/L. Although 25(OH)D concentration was inversely associated with death from cardiovascular disease, cancer and other causes, the authors noted that current scientific evidence does not support any benefit of vitamin D supplementation against these diseases. Independently, Lindqvist et al. [9] estimated that sun avoidance was a risk factor for death of similar magnitude to smoking. Garland et al. [10] reported that people with serum 25(OH)D <22 nmol/L had nearly twice the age-adjusted death rate compared with those with >125 nmol/L. A Mendelian randomization analysis by Afzal et al. [11] showed that genotypes associated with low serum 25(OH)D were associated with a 14% increase in all-cause mortality for every decline of 20 nmol/L, but not an increase in cardiovascular disease mortality. Overall, these results suggest that low 25(OH)D is associated with increased mortality, as indicated in the meta-analyses [8,10].

### 2.2. Internal Cancers

#### 2.2.1. Breast Cancer

McDonnell et al. [12] found that a 400% increased risk of breast cancer was associated with serum 25(OH)D <50 nmol/L compared to >150 nmol/L, and that there was a dose-response relationship between 100 to 150 nmol/L and 125 to 174 nmol/L. Mohr et al. [13] found that, in patients with breast cancer, a 79% decreased risk of death was associated with serum 25(OH)D levels in the highest quintile relative to the lowest quintile in five studies (>72 nmol/L vs. <50 nmol/L, >55 nmol/L vs. <35 nmol/L, >81 nmol/L vs. <46 nmol/L, >75 nmol/L vs. <75 nmol/L, >75 nmol/L vs. <50 nmol/L). However, a recent Mendelian randomization analysis found no evidence to support a causal association between 25(OH)D and risk of breast cancer [14].

#### 2.2.2. Colorectal Cancer

Gorham et al. [15], examined five studies on association of serum 25(OH)D and colorectal cancer risk. A meta-analysis indicated a 104% higher risk associated with serum 25(OH)D <30 nmol/L compared to >82 nmol/L. Rebel et al. [16] reported a small carcinogenesis study in genetically modified mice, which showed reduced colorectal cancer load (area covered by tumors) in animals supplemented with vitamin D or exposed to UVR. Neither vitamin D nor UVR was associated with tumor numbers. However, UVR but not dietary vitamin D appeared to reduce progression to malignancy.

### 2.3. Cardiovascular Disease

Cardiovascular disease has been the leading cause of years of life lost globally for the last three decades [17], and hypertension, the leading risk factor for cardiovascular and cerebrovascular disease, underlies 18% of all deaths worldwide [18]. A growing body of evidence shows an inverse relationship between sunlight exposure, and blood pressure and cardiovascular disease [19].

Nitric oxide (NO) is a ubiquitous signaling molecule and an important endogenous vasodilator produced by the vascular endothelium (innermost layer of blood vessels) [20]. This knowledge has changed our understanding of hypertension because it shows that high blood pressure can develop not only as a result of an overproduction of vasoconstrictor substances such as angiotensin and adrenalin, but also as a consequence of impaired synthesis production of a continuously produced vasodilator substance such as NO [20]. In addition to acting as a signaling molecule, NO is a potent antioxidant capable of modulating whole body redox status [21,22,23].

Skin contains significant amounts of preformed storage forms of NO [24]. The origin of NO precursors in skin, which include nitrate (NO_3_^−^), nitrite (NO_2_^−^) and nitroso compounds (RXNO), is unclear; they may be derived from bacterial conversion of sweat constituents or actively produced by innate mechanisms. Feelisch et al. [25] hypothesized that these chemical species in skin may be mobilized into the circulation after UVR exposure, thus expanding their action radius from local to systemic.

Studies in healthy humans [26] showed that a 20 J/cm^2^ UVA exposure (equivalent to ~30 min of mid-day Mediterranean sun and suberythemal in light skin) relaxes arterial resistance in association with NO release. Exposure of the body to UVA lowered blood pressure independently of temperature and serum 25(OH)D. This effect was associated with increased plasma concentrations of nitrite (a relatively stable oxidation product of NO) and lower levels of nitrate. Most ‘NO release activity’ was found in the upper epidermis [26]. These findings are important because hypertension is the leading cause of non-communicable diseases despite current pharmacotherapy [27]. A recent observational study, in a large cohort of chronic hemodialysis patients, confirmed that environmental UVR exposure is inversely associated with blood pressure independently of ambient temperature [23]. These results are in line with the Mendelian randomization findings that genotypes associated with low serum 25(OH)D are associated with increased all-cause mortality but not with cardiovascular mortality. This supports the view that a mediator other than vitamin D contributes to the observed reduction in cardiovascular mortality [11]. In line with the above hypothesis is the finding that those with habitual low sun exposure were at twice the risk of cardiovascular mortality compared with those with greatest sun exposure [9] and that daytime myocardial infarctions reduce with increased sunlight in summer [28].

### 2.4. Metabolic Syndrome

Geldenhuys et al. [29] found that UVR suppressed obesity and type 2 diabetes in a murine model, but these benefits were not reproduced by vitamin D supplementation. UVR suppression of metabolic syndrome development was blocked by application of a NO scavenger to the skin and reproduced by application of a topical NO donor cream [29]. These results complement experimental evidence for the involvement of NO in metabolic syndrome obtained with genetic models [30] and suggest that sunlight exposure may be an effective means of suppressing the development of obesity and metabolic syndrome through vitamin D-independent mechanisms. A study nested in the Rotterdam Study, a population-based cohort of middle-aged and elderly adults, found that serum 25(OH)D <50 nmol/L was associated with a 64% increased risk of metabolic syndrome relative to >75 nmol/L, and that lower serum 25(OH)D was still significantly associated with higher odds of metabolic syndrome after adjustment for body mass index (BMI) [31]. Afzal et al. [32] measured serum 25(OH)D in 9841 people, of whom 810 developed type 2 diabetes during 29 years of follow up. The investigators observed an association of low serum 25(OH)D with a 35% increased risk of type 2 diabetes for the lowest (<12 nmol/L) relative to the highest (>50 nmol/L) quartile of 25(OH)D.

### 2.5. Neurological Conditions

#### 2.5.1. Alzheimer’s Disease and Other Cognitive Decline

Littlejohns et al. [33] measured serum 25(OH)D in 1658 persons, of whom 171 developed dementia from any cause, including 102 cases of Alzheimer’s disease during a mean follow up of 5.6 years, and found a 125% increased risk for all-cause dementia and a 122% increased risk for Alzheimer’s disease in people with serum 25(OH)D <25 nmol/L compared to >50 nmol/L. Keeney et al. [34] found that dietary vitamin D deficiency in rats contributes to significant direct and indirect oxidative stress in the brain and might promote cognitive decline in middle-aged and elderly adult rats. These findings are in line with the assumption of a link between redox and vitamin D status in the brain [34] and perhaps even at the whole body level.

#### 2.5.2. Autism

Autism spectrum disorder (ASD) is a heterogeneous group of neurodevelopmental conditions characterized by repetitive behaviors and deficits in social relationships. While largely explained by genetic variants, environmental factors probably contribute to the onset of the disorder. These include prenatal and early life exposures such as infection, obstetric complications and nutritional or toxin-related exposures. There is growing interest in the possible links between gestational vitamin D deficiency and the risk of ASD [35,36]. Birth cohort studies have provided evidence that prenatal vitamin D deficiency is associated with a range of later neurodevelopmental outcomes, including impaired language and cognitive development in offspring [37,38]. Vinkhuyzen et al. [39] found, in neonates and pregnant mothers at midgestation, that serum 25(OH)D concentrations of <25 nmol/L relative to >50 nmol/L were correlated with a 142% increased risk of autism in the child. There are mechanistic studies linking vitamin D deficiency and related metabolic processes to abnormal brain development. The active form of vitamin D (1,25(OH)D) is known to affect the function of voltage-gated calcium channels. Variants in genes coding for subunits of these same calcium channels (e.g., *CACNA1C*) are associated with the risk of both schizophrenia and ASD [40]. However, a prospective community trial of vitamin D supplementation combined with lifestyle advice is needed to test interventions for autism and other neurodevelopment disorders.

### 2.6. Asthma, Respiratory Infection and Allergy

Morgan et al. [41] noted that asthma incidence has increased in parallel with temporal increases in vitamin D deficiency, that epidemiological studies have reported associations between low serum 25(OH)D and increased asthma incidence, and that some evidence exists for a positive correlation between asthma incidence and latitude. Hollams et al. [42] reported that serum 25(OH)D <50 nmol/L in childhood was positively associated with increased risk for asthma or wheeze in a small longitudinal study of 263 children (198 participated up to age five, and 147 up to age 10). Zosky et al. [43] found that maternal serum 25(OH)D <50 nmol/L at 16–18 weeks compared to >75 nmol/L was associated with reduced lung function in children at age six.

Although trials of vitamin D supplementation have shown limited benefits [44], one analysis of 25 trials has shown a favourable outcome for respiratory tract infection. This suggested that daily vitamin D supplementation ranging from <400 IU (10 μg) to >2000 IU (50 μg) reduced the relative risk of development of respiratory tract infections by 12% [45]. This effect was stronger when starting serum 25(OH)D concentration was low. Asthma development generally reflects respiratory infection in allergen-sensitized individuals; it has been reported that vitamin D supplementation can reduce exacerbations of asthma that require emergency treatment or administration of systemic corticosteroids [46]. However, this benefit was independent of starting 25(OH)D levels. Mechanistically, vitamin D may not only regulate immune cell activity to reduce infections and sensitization but may also regulate the development of structural cells of the lung [41]. Some trials have also reported reduced development of wheeze or asthma, but not respiratory tract infections, in offspring of pregnant women supplemented with vitamin D; this effect was independent of starting 25(OH)D levels. The outcomes of trials of vitamin D supplementation to prevent allergic sensitization in infants are, however, controversial, with some trials reporting no significant benefits [47].

Associations between offspring lung development and maternal UVR exposures at 18 weeks of pregnancy have been reported in an Australian cohort study but vitamin D-dependent and independent mechanisms could not be dissected [48]. They were, however, dissected in a study of eczema development. Vitamin D supplementation (400 IU (10 μg)/daily for six months) did not affect eczema development in children of highly allergic mothers [49]. However, this rate was reduced by ~50% in children exposed to higher UVR doses. The effect was also extracutaneous as peripheral blood mononuclear cells from those children receiving higher UVR exposures also produced lower levels of inflammatory cytokines.

### 2.7. Autoimmune Diseases

#### 2.7.1. Type 1 Diabetes

The pathogenesis of type 1 diabetes (T1D)—one of the few chronic diseases that begin in childhood—is not entirely known, but both genetic and environmental factors are associated with the disease risk. The global increase in the incidence of T1D during the past decades [50] suggests a contribution from environmental risk factors, of which vitamin D deficiency due to insufficient sun exposure might be one [51]. Furthermore, seasonality of birth in T1D cases [52] and documented birth cohort effects in T1D incidence [53] suggest that lack of sunshine during gestation or in early infancy may be a risk factor for T1D. Jacobsen et al. [54] studied a Danish population of 331,623 individuals and found that less ambient sunlight (below the median compared to above the median) during the third trimester of gestation was associated with a 67% higher risk of T1D in males aged five to nine years.

#### 2.7.2. Multiple Sclerosis

Multiple sclerosis (MS) is an autoimmune disease of the central nervous system characterized by demyelination and loss of axonal function. Low serum 25(OH)D and low sun exposure have received considerable attention as possible causes of MS. Since sun exposure is the major determinant of 25(OH)D in many populations, 25(OH)D concentration has been suggested as a marker of sun exposure [55].

Evidence from prospective studies is relatively consistent in showing that serum 25(OH)D >100 nmol/L is associated with a 50–60% decreased risk of MS compared with <50 nmol/L [56,57], while 25(OH)D <30 nmol/L relative to >50 nmol/L is associated with a two-fold increase in MS risk [58]. Low prenatal 25(OH)D may increase the risk of developing MS in adulthood [59,60], but this is not consistently found [61].

There have been studies of past sun exposure and MS risk. In Tasmania, Australia (41–44 ^o^S), recollection of having spent at least 4 h in the sun daily (vs. <1 h in winter or <2 h in summer) between the ages of six and 10 years was associated with a 50% reduced risk of MS [62]. Studies in Norway and Italy [63], Sweden [64] and the US [65] reached similar conclusions, and Langer-Gould et al. [66] have corroborated these findings. Importantly, given the difficulty with accurate recall of past sun exposure, Lucas et al. [67] measured solar damage on the back of the hand, an objective measure of lifetime sun exposure, and also demonstrated low sun exposure may be an important MS risk factor.

There is some evidence that past sun exposure and serum 25(OH)D concentration may be independent risk factors for MS [66,67], suggesting that sunlight may modulate MS risk through vitamin D and non-vitamin D pathways [68,69,70]. Mendelian randomization studies support a causal role for vitamin D [69,70], while a multiethnic study found a reduced risk of MS or clinically isolated syndrome (the earliest detectable form of MS) with higher 25(OH)D levels in whites (Fitzpatrick skin types (FST) I–III) but not blacks (FST V–VI) or Hispanics (FST IV), with a more consistent effect across all ethnic groups for higher past sun exposure [66].

Recent meta-analyses of vitamin D supplementation trials show, variously, no effect on MS outcomes [70,71], a beneficial effect for some (mainly non-clinical) outcomes, or a suggestion of an adverse effect of high-dose supplementation. One small randomized clinical trial has studied the effects of narrowband UVB (~311 nm) phototherapy on patients with clinically isolated syndrome. Despite a 30% reduction in conversion to MS, the results were not statistically significant [72].

### 2.8. Myopia

Myopia (short-sightedness) is an increasing global health problem, especially in East and South East Asia [3]. In addition to requiring corrective lenses, myopia poses an increased risk of retinal detachment and blindness. A meta-analysis of five randomized controlled trials in China and Singapore (n = 3014) concluded that time outdoors reduced the risk of myopia [73]. A multi-European country study concluded that increased estimated UVB exposure was associated with reduced myopia, especially in childhood and young adulthood [74]. UVB was used as a proxy for solar exposure and there was no evidence of a relationship between vitamin D status and myopia. In general, the evidence for a role for vitamin D in myopia is mixed [3]. One study in China concluded that an additional 40 min of outdoor activity at the end of the school day, as well as weekend outdoor activities, reduced the cumulative incidence rate of myopia over three years [75]. A review by French et al. [76] stated that epidemiological evidence suggests that children who spend more time outdoors are less likely to be or to become myopic, irrespective of how much near work they do or whether their parents are myopic. The likely mechanism for this protective effect is visible light (400–700 nm) stimulating release of dopamine from the retina, which inhibits increased axial elongation, the structural basis of myopia. The authors describe the effect of time outdoors on the risk of myopia as robust. Recommending increased outdoor exposure to prevent myopia must be mitigated by considering its adverse effects.

## 3. Evidence for Harms to Health from Sun Exposure

### 3.1. Skin Cancer

Melanomas originate from melanocytes, which are the pigment-producing cells in the epidermis. Squamous cell carcinoma (SCC) and basal cell carcinoma (BCC) originate from keratinocytes that are the majority epidermal cell type. The majority of skin cancers are keratinocyte cancers, but melanoma is the main cause of death from skin cancer. In its most recent expert review of the evidence, the International Agency for Research on Cancer concluded: “solar radiation causes cutaneous malignant melanoma, squamous cell carcinoma of the skin and basal cell carcinoma of the skin” [77]. There is, therefore, little or no doubt that solar exposure is the major cause of cutaneous melanoma and keratinocyte cancers.

#### 3.1.1. Melanoma

Epidemiological studies identify sunburn as a strong risk factor for melanoma [3]. Five sunburns per decade vs. no sunburn showed a relative risk (RR) of 3.24 (95% confidence interval (CI), 2.19–4.66) [78]. Another study showed a RR of 1.83 (95% CI 1.59–2.12) for many sunburns vs. few [79]. These observations are congruent with the consistent and moderately strong association that is observed between intermittent sun exposure (mostly recreational sun exposure) and melanoma; a meta-analysis of 33 studies showed a RR of 1.61 (95% CI 1.31–1.99) [80].

More continuous (chronic) sun exposure, on the other hand, appears to have a null or an inverse association with melanoma. The meta-analysis cited above found a pooled RR of 0.95 (95% CI 0.87–1.04). Some more recent studies reported: an RR of 1.22 (95% CI 0.82–1.81) for the highest relative to the lowest quartile of weekday sun exposure (Australian Melanoma Family Study, [81]); 0.78 (95% CI 0.61–1.01) for the highest relative to the lowest quartile (Genes, Environment and Melanoma Study, [81]); and 0.91 (95% CI 0.81–1.01) for high relative to low continuous sun exposure [79]. It is important to note that in almost all these studies the reference groups were those with low chronic exposure to the sun, not low total exposure to the sun. People with high intermittent sun exposure or sunburn would often be in the low chronic exposure category and could therefore mask a positive association of chronic sun exposure with melanoma. In principle, this issue could be addressed analytically, but inaccuracies in recall of the various sun exposure variables would be likely to threaten the validity of the analysis results.

There is also a class of melanoma, lentigo maligna melanoma (LMM), which comprises 4–15% of melanoma cases [82] and appears mostly on sun-damaged areas of the face and neck of older persons but has not been associated with either sunburn or chronic sun exposure in epidemiological studies [78,79,83,84]. However, since LMM appears mostly on sun-damaged skin, a linkage with sun exposure is highly likely. There are few epidemiological studies examining the association of sun exposure with LMM and, because of LMM’s comparative rarity, the number of cases of LMM examined in such studies is small and the studies inconclusive [78,79,83,84].

There is evidence of biological mechanisms whereby sunburn may be particularly important in causing some melanomas. Epidermal melanocytes are generally non-replicating cells [85,86,87], and it may be that sunburn or other trauma is required to stimulate melanocyte replication [88]. It is only when cells divide that unrepaired or misrepaired DNA photodamage can be translated into potentially cancer-causing mutations. Sunburn in mice generates a proliferative response in follicular melanocyte stem cells, the progeny of which migrate into the interfollicular epidermis where melanomas arise [89]. While this evidence indicates that some mechanisms of melanoma genesis may require sunburn to initiate them, it does not exclude other mechanisms.

#### 3.1.2. Keratinocyte Cancers

Like melanoma, epidemiological studies clearly show that sunburn increases the risk of SCC: odds ratio (OR) 2.32; 95% CI 1.46–3.70 for six to 10 sunburns in childhood [90] and OR 1.23; 95% CI 0.90–1.69 for sunburn at any age [82]. Sunburns also increase the risk of BCC: OR 2.42; 95% CI 1.74–3.36 for more than three lifetime sunburns vs. none [91], OR 2.33; 95% CI 1.62–3.36 for six to 10 burns in childhood [92], OR 1.40; 95% CI 1.29–1.51 for sunburn at any age [82]. In contrast to melanoma, epidemiological studies also show that chronic sun exposure increases the risk for SCC: OR 6.5; 95% CI 1.7–25.6 for >40,000 lifetime hours of sun exposure relative to no significant increased risk for <20,000 lifetime hours for residents of the Netherlands [93] and OR 8.0 (no CI figures published) for 200,000 lifetime hours of sun exposure relative to no significantly increased risk for <70,000 lifetime hours for residents of southern Europe [92]. Chronic sun exposure also increases the risk for BCC: OR 2.3; 95% CI 0.96–5.7 for >40,000 lifetime hours of sun exposure relative to <20,000 lifetime hours in residents of the Netherlands [93], and OR 2.0 (no CI figures published) for lower exposures (8000–10,000 lifetime hours) with a plateau and a slight decrease of risk for the highest exposures (200,000 lifetime hours) for residents of southern Europe [92].

### 3.2. Photoprotection of Skin by Sunscreens

Studies have shown that sunscreens are not particularly effective in preventing sunburn outside laboratory conditions, probably because they are not used properly [94]. There is evidence that regular sunscreen use can inhibit melanoma and SCC but not BCC [95,96,97]. There has been concern that sunscreens, which must attenuate UVB to prevent sunburn, may have an adverse effect on cutaneous vitamin D synthesis. However, two recent reviews have concluded that sunscreen use has little or no impact on vitamin D status [98,99]. Narbutt et al. [100] and Young et al. [101] showed not only that proper use of sunscreen (i.e., ≥2 mg/cm^2^) prevents sunburn, but that it can do so while not unduly interfering with vitamin D synthesis. In these studies, 62 Polish volunteers on a week-long sunny holiday in March on the Canary Islands (approximately same latitude as Florida) were divided into two groups: 22 applied their own sunscreen without instruction and 40 were given sun protection factor (SPF) 15 sunscreen and guidance on optimal application. Individuals who followed the effective application guidelines had no sunburn on five exposed body sites during seven cloudless days while those who applied their own sunscreen had daily sunburn on all sites. Both groups had significant increases in serum 25(OH)D concentrations.

### 3.3. Adverse Effects on the Eye

Sight depends on visible radiation reaching the retina. The cornea and intraocular lens prevent UVR reaching the retina in adults, but some UVR is transmitted in children. A description of the adverse effects of UVR on the eye is given by Lucas et al. [3,102]. In summary, exposure of the eye to sunlight causes cataract, photoconjunctivitis, pterygium macular and retinal degeneration [103,104] and possibly conjunctival and uveal melanoma (UM) [77]. A recent genetic analysis of UM showed UVR signature mutations (cytosine (C) to thymine (T) transition) in iris melanomas [105] but no other UM. It is not known if these are passenger or driver mutations. The strongest evidence in relation to cataract is for cortical cataract, with weaker evidence for associations with nuclear and posterior subcapsular cataract [102]. Although cataract is very common and an important cause of blindness where access to cataract surgery is limited, cortical cataract is the least likely of the cataract types to cause blindness.

## 4. Conclusions

Most research on the health effects of sun has focused on skin cancer. Many ecologic studies that show an inverse relationship between vitamin D status and a wide range of diseases have suggested beneficial health effects from solar exposure that have been widely attributed to vitamin D. However, supplementation studies have, in general, not supported vitamin D as causal. This raises the possibility that poor vitamin D status is a proxy for insufficient sun exposure that impacts on health and is therefore a biomarker of other mediators. These are not established but nitric oxide released from skin is a strong candidate that merits further research. It is also possible that sunlight may have direct systemic effects via blood borne cells. The studies discussed in this paper suggest that it would be wise for public health bodies to have a better understanding of the risks and benefits of solar exposure, which will depend on Fitzpatrick skin type and local UVR conditions.

Current guidelines for solar UVR exposure are based on vitamin D synthesis. For example, Polish authorities suggest at least 15 min per day between 10.00 and 15.00 h for adolescents and adults between May and September [106]. A study based on a UK climatology model recommends 9–13 min (depending on latitude) lunchtime exposure between March and September in season-appropriate clothing [107]. Exposure times should be increased with darker FST but recent studies suggest that the inhibitory effect of melanin on vitamin D synthesis is relatively small compared with erythema [108]. A maximum of 30 min unprotected exposure is advised. For vitamin D at least, the UVR doses necessary for its synthesis are much lower than for sunburn [101], and exposure time should be guided by the local UVI that is readily available online. There is no advantage with excessive exposure as a complex of photochemical reactions limits the production of pre-vitamin D that reaches a maximal level in a relatively short time [109]. It is important that eyes are protected while sunbathing, by keeping them closed or wearing sunglasses and/or contact lenses that filter UVR [103].

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
