# Peer review of "Insufficient Sun Exposure Has Become a Real Public Health Problem"

_ijerph, 2020, doi:10.3390/ijerph17145014_

Round 1

Reviewer 1 Report

The authors present an interesting and important perspective regarding sun exposure. However, I have several concerns regarding the structure of this manuscript.

In general, I would like to see a more cohesive description of the health benefits of sun exposure rather than simply a listing of diseases and supporting literature.

There is little discussion of the alternatives to vitamin D synthesis as the primary beneficial effect, although it is clear the authors support other beneficial mechanisms of sun exposure rather than vitamin D supplementation.

Individual disease topics do not all provide studies regarding vitamin D supplementation for potential to prevent the conditions. For some, no studies are cited regarding trials of vitamin D supplementation. Given the topic and stated longstanding reliance on vitamin D as the most important mechanism for the benefits of sun exposure, the presence or absence of studies for each disease should be clearly stated and discussed.

Some sections are reasonably well written, such as the ones discussing multiple sclerosis and melanoma. Simply adding an introductory sentence to begin each section would improve flow. Development of a structure to follow that is the same for each section would be quite beneficial.

Lines 40-47: I have no doubt the authors are indeed experts in their fields and have conducted high quality primary research throughout their careers. Although it is stated this is not intended as a “systematic review” it should be presented as at least a mini non-systematic review. As written this manuscript lacks cohesiveness and flow, as though it were written in pieces by different authors without regard for readability. It is lacking good flow and transition statements throughout. Please revise.

Lines 71-74: Sentence structure is very poor and difficult to understand. Not a clearly argued point.

Lines 220-228: The description of myopia is particularly weak as only one review paper is cited. Given the other sections are much more detailed with several primary studies cited, a similar level of diligence for sun exposure as protective against myopia should be pursued and cite several primary research papers. Numerous epidemiologic and clinical trials are published on this topic.

Lines 282-287: This is a very brief discussion given the large body of work on adverse effects of sun exposure on the eye. Expanding this section to at least comparable in length to the other topics discussed would strengthen the manuscript. Prior to the relatively recent discovery that sun exposure was protective against myopia there were no known ocular benefits.

Author Response

In adequate sun exposure has become a real public health problem
Manuscript ID: ijerph-839833

Overall response: We thank the reviewers for their constructive comments, and we have used these to improve the manuscript. We have also tightened the text and added a few recent references that were not published at the time of the meeting. Minor editorial changes to words/sentence construction are not highlighted in the revision, but more substantial changes are shown in purple text. Less than and more than have been replaced by < and > respectively and serum 25(OH)D is now presented as nmol/L. The conclusions have been re-written. There are some minor changes to the abstract.
It should be noted that reviewers 1 & 2 disagree on the structure of the paper and reviewer 3 makes no comment on this. We have tried to improve the structure and flow in response to reviewer 1.

Reviewer 1
Comments and Suggestions for Authors
The authors present an interesting and important perspective regarding sun exposure. However, I have several concerns regarding the structure of this manuscript.
1. In general, I would like to see a more cohesive description of the health benefits of sun exposure rather than simply a listing of diseases and supporting literature.
Response: It would be difficult to give any detail without sub-division of the diseases. However, some general statements on sunlight and the effects of sun on health have been given in the introduction. There is also a new introduction to the skin cancer section. A few words on context have been added to sections where appropriate.
2. There is little discussion of the alternatives to vitamin D synthesis as the primary beneficial effect, although it is clear the authors support other beneficial mechanisms of sun exposure rather than vitamin D supplementation.
Response: The reviewer makes a good point. We lack data on the possible mediators apart from vitamin D and nitric oxide (NO). We have added 3 recent papers on the effect of ultraviolet radiation (UVR) on the skin and blood transcriptome. Many genes are up- or down-regulated and the consequences of this are largely unknown, especially at the systemic level. What is interesting is that the studies on the blood transcriptome show that genes relating to immunity are affected. One of the papers on the blood transcriptome reported that the effect was independent of change of vitamin D status. This raises the possibility that UVR may be having a direct effect on immunity via white blood cells.
3. Individual disease topics do not all provide studies regarding vitamin D supplementation for potential to prevent the conditions. For some, no studies are cited regarding trials of vitamin D supplementation. Given the topic and stated longstanding reliance on vitamin D as the most important mechanism for the benefits of sun exposure, the presence or absence of studies for each disease should be clearly stated and discussed.
Response: The papers cited reflect the discussion at the meeting and was not intended to be comprehensive. However, some sections have been expanded and some new relevant papers (i.e. since the meeting) have also been added.
4. Some sections are reasonably well written, such as the ones discussing multiple sclerosis and melanoma. Simply adding an introductory sentence to begin each section would improve flow. Development of a structure to follow that is the same for each section would be quite beneficial.
Response: The flow has been improved as suggested. Introductory sentences have been added where appropriate.
5. Lines 40-47: I have no doubt the authors are indeed experts in their fields and have conducted high quality primary research throughout their careers. Although it is stated this is not intended as a “systematic review” it should be presented as at least a mini non-systematic review. As written this manuscript lacks cohesiveness and flow, as though it were written in pieces by different authors without regard for readability. It is lacking good flow and transition statements throughout. Please revise.
Response: We have changed the text on the “review” as suggested by this reviewer. One author (ARY) has revised the whole text to ensure an even style throughout.
6. Lines 71-74: Sentence structure is very poor and difficult to understand. Not a clearly argued point.
Response: This sentence has been re-written.
7. Lines 220-228: The description of myopia is particularly weak as only one review paper is cited. Given the other sections are much more detailed with several primary studies cited, a similar level of diligence for sun exposure as protective against myopia should be pursued and cite several primary research papers. Numerous epidemiologic and clinical trials are published on this topic.
Response: This reflects the content of the meeting, but this section has been expanded with recent work and additional references.
8. Lines 282-287: This is a very brief discussion given the large body of work on adverse effects of sun exposure on the eye. Expanding this section to at least comparable in length to the other topics discussed would strengthen the manuscript. Prior to the relatively recent discovery that sun exposure was protective against myopia there were no known ocular benefits.
Response: Again, this reflects the discussion at the meeting. This section has been expanded slightly with additional references.

Reviewer 2 Report

The paper „Inadequate sun exposure has become a real public health problem” by Alfredsson et al. is well organised and has a clear construction. The work reviews current knowledge of UV radiation impact on human health, especially positive role of solar radiation. Although the work is not original, it is still very much needed to make us aware of the importance of solar exposure, especially that some effects cannot be replaced by supplementation of vitamin D.

General comments:

The paper is well written, with clear construction. The abstract summarize the work adequately, describes the overall work and contains conclusions. The problem raised by the researches is of a great importance. The article should be published but can be improved. The reviewer has a few detailed comments, which may be helpful to the authors. Also the document should be edited, because it contains many multiple spaces.

Detailed comments:

Page 2, line 53: „Hence, in the following summaries higher serum vitamin D concentrations (measured as 25-hydroxy vitamin D [25(OH)D]) are considered to be a proxy for sun exposure so that their association with health benefits will be taken as indicating a beneficial effect of sun exposure and not necessarily a benefit of vitamin D.” - I am not sure if this assumption is correct. What about individuals, that supplement vitamin D, thus they have correct vitamin D status, but do not expose themselves to the solar radiation?

Page 3, line 112: Is there any action spectrum or threshold dose for this effect? Does „30 minutes of midday sunshine in the Mediterranean” refer to any phototype?

Page 5, line 201: „In Australia, having spent at least 4 hours (vs. <1 in the winter or <2 in summer) in the sun daily between the ages of 6 and 10 was associated with a 50% reduced risk of MS [47]” - what authors mean by „(vs. <1 in the winter or <2 in summer)”? Could you explain?

Page 5, line 212: „with higher 25(OH)D levels in whites but not blacks or Hispanics” - try to use more scientific language in relation to skin phototypes.

Page 6, line 235: „Epidemiological studies show that sunburns are strong risk factors for melanoma: RR 3.24; 95% CI, 2.19-4.66 for five sunburns per decade of life vs. no sunburns [61]; RR 1.83; 95% CI, 1.59-2.12 for many sunburns vs. few [62].” - please explain shortcuts „RR” and „CI” and give a short explanation to this sentence, because it is unclear.

Page 7, line 288: authors should cite also those studies, that confirm that sunscreens indeed block vitamin D skin-synthesis. My overall comment is that: the vitamin D status for those that regularly use sunscreens improves as the effect of spending more time outdoors (not because sunscreens do not partially block vitamin D synthesis). Further, in the cited reference, volunteers used SPF 15, but dermatologists recommend using at least SPF 30.

Page 7, lines 306-311: when giving guidelines, authors should mention exposure ratio (ER), because UV radiation has different effects in the different parts of the body. If the horizontally oriented surfaces are covered, than the time of exposure could be longer. Also, the presented concept is nothing new. Referring to the newest guidelines of Poland and UK for vitamin D supplementation, it is recommended to spend some time outdoors without photoprotection.

References:

A. Rusińska, P. Płudowski, M. Walczak, M.K. Borszewska-Kornacka, A. Bossowski, D. Chlebna-Sokoł, J. Czech-Kowalska, A. Dobrzańska, E. Franek, E. Helwich, T. Jackowska, M.A. Kalina, J. Konstantynowicz, J. Książyk, A. Lewiński, J. Łukaszkiewicz, E. Marcinowska-Suchowierska, A. Mazur, I. Michałus, J. Peregud-Pogorzelski, H. Romanowska, M. Ruchała, P. Socha, M. Szalecki, M. Wielgoś, D. Zwolińska, A. Zygmunt, Vitamin D supplementation guidelines for general population and groups at risk of vitamin D deficiency in Poland-recommendations of the polish society of pediatric endocrinology and diabetes and the expert panel with participation of national specialist consultants and representatives of scientific societies-2018 update, Front Endocrinol (Lausanne) 31 (9) (2018) 246, https://doi.org/10.3389/fendo.2018.00246.

East Lancashire Health Economy Medicine Management Board (ELMMB) http://www.elmmb.nhs.uk

Scottish Government https://www.gov.scot/publications/vitamin-d-advice-for-allage-groups

Author Response

Reviewer 3
The paper entitled “Inadequate sun exposure has become a real public health problem”. This article aims to alert the medical community and public health authorities to accumulating evidence on the health benefits of sun exposure, which suggests that inadequate sun exposure is a significant public health problem.
This work is interesting but unfortunately, the manuscript is not properly prepared for the Journal. Probably the authors have not read the “instruction for the authors”. The entire manuscript should be corrected.
1. It is mandatory to correct the manuscript in some points:
Response: This has been done as suggested by the reviewers above and the text language tightened.
2. Please complete affiliation.
Response: The affiliations have been added.
3. Line 13, 16, 19, 20: The author should remove double spaces.
Response: The text has been revised according to the requirements of the journal.
4. Read the full article and remove double spaces.
Response: The text has been revised according to the requirements of the journal.
5. Line 22: define: UVR
Response: This has been done and expanded in the 1st paragraph of the introduction.
6. Line 40-47 – edit paragraph.
Response: This paragraph has been edited
7. Put a comma after the cited name, e.g. line 59: should be “Chowdhury et al. [3], ….” – correct the entire manuscript.
Response: This has been done
8. Line 112: define “UVA”
Response: This has been done in the introduction and specific wavelengths given where available.
9. Line 191: Remove the MS shortcut and use the entire name (in all article).
Response: We feel that MS is appropriate because it is a widely used abbreviation for multiple sclerosis. Furthermore, it is used multiple times in the relevant section.
10. Line 217: define “UVB”
Response: This has been done in the introduction and specific wavelengths given where available. In this case it is near monochromatic radiation of ~311nm
11. Line 235: should be: “Epidemiological studies show that sunburns are strong risk factors for melanoma: relative risk (RR) 3.24;” – Right?
Response: This section has been re-written at the request of reviewer 2
12. References: All references must be corrected. Read the instructions for the authors.
Response: The references have been reviewed and have been listed in the journal style using EndNote.

Reviewer 3 Report

Comments and Suggestions for Authors

The paper entitled “Inadequate sun exposure has become a real public health problem”. This article aims to alert the medical community and public health authorities to accumulating evidence on the health benefits of sun exposure, which suggests that inadequate sun exposure is a significant public health problem.

This work is interesting but unfortunately, the manuscript is not properly prepared for the Journal. Probably the authors have not read the “instruction for the authors”. The entire manuscript should be corrected.

It is mandatory to correct the manuscript in some points:

# Please complete affiliation.

# Line 13, 16, 19, 20: The author should removed double spaces.

# Read the full article and remove double spaces.

# Line 22: diefine: UVR

# line 40-47 – edit paragraph.

# Put a comma after the cited name, e.g. line 59: should be “Chowdhury et al. [3], ….” – correct the entire manuscript.

# line 112: define “UVA”

# line 191: Remove the MS“ shortcut and use the entire name (in all article).

# line 217: define “UVB”

# line 235: should be: “Epidemiological studies show that sunburns are strong risk factors for melanoma: relative risk (RR) 3.24;” – Right?

# References: All references must be corrected. Read the instructions for the authors.

Author Response

(The authors gave the same response as above.)

Round 2

Reviewer 1 Report

This manuscript highlights a potentially significant public health concern that disputes some commonly held beliefs among the general population and scientists. The revised version of this manuscript is substantially improved in both content and flow. The authors have adequately addressed my previous concerns.

Reviewer 3 Report

Authors changes improved  the manuscript. The change of title is very appropriate.